# Reinforcement Learning Agents in Quantum Code Discovery with Argmax-Preserving Quantization

## Abstract

Reinforcement learning (RL) has recently been employed to autonomously discover quantum error-correcting codes and encoders tailored to specific noise models and hardware constraints. In these settings, the policy operates over a large discrete action space, and low-bit quantization of policy logits can perturb action rankings—especially when action-value margins are small—thereby altering exploration trajectories and degrading the quality of discovered codes. We propose *Argmax-Preserving Quantization (APQ)*, a quantization-aware training framework that explicitly regularizes the action ranking of a quantized policy with respect to a full-precision teacher. APQ minimizes ranking errors between full-precision and low-bit policies to preserve the top-1 action under INT8 representations, and augments this with a reward-safe constraint that bounds perturbations of Knill–Laflamme–based code-quality metrics under quantization. Experiments with policy-gradient agents on Clifford-simulated environments show that APQ enables INT8 networks to discover $[[n, k, d]]$ stabilizer codes with distance up to $d = 5$ and logical error suppression comparable to FP16 baselines, while reducing policy-inference cost by a factor of 3.8. These results indicate that decision-consistent quantization can substantially accelerate RL-based quantum code discovery without sacrificing code quality, and provide a general recipe for stabilizing discrete-action RL policies under low-bit quantization.

## 1 Introduction

Reinforcement learning (RL) has recently emerged as a promising engine for *automated* discovery of quantum error-correcting codes (QECCs) and encoders under hardware and noise constraints. Beyond optimizing decoders, recent work shows that RL agents can discover both codes and encoding circuits with rewards grounded in the Knill–Laflamme (KL) conditions and scalable Clifford simulators, reaching nontrivial distances (e.g., $d \leq 5$) and tailoring to device-specific noise (Olle et al., 2024; Su et al., 2025; He & Liu, 2025). This trend aligns with a broader push to close the loop between algorithmic search and near-term hardware by exploiting efficiently simulable stabilizer or Clifford environments (Gidney, 2021).

However, deploying such agents efficiently on real systems exposes a gap: the computational bottleneck of policy evaluation and selection within long RL training runs. In RL-based QEC code discovery, each episode combines expensive Clifford simulation and repeated policy queries over a large discrete action space (gate insertions, rewrites, and termination decisions), so even constant-factor improvements in policy inference can translate into substantial wall-clock savings. Quantization of the policy network is a standard route to reduce inference latency and energy, and has been shown to accelerate deep RL training and deployment (e.g., INT8 actors) without harming convergence in several settings (Krishnan et al., 2022). In our setting, this amounts to applying low-bit quantization to the policy logits. Yet, when the action space is large and many actions are nearly tied in value, such quantization can perturb *decision orderings*: small changes to logits or Q-values may flip the $\arg\max$ and thereby alter exploration trajectories and long-horizon returns. Recent quantization studies outside RL similarly note that preserving ranking/ordering (e.g., attention-score order) is critical for decision fidelity, motivating ranking-aware objectives during quantization (Fu et al.,

2025). In QEC code discovery, these mis-rankings correspond to different gate sequences and encoder topologies, which can amplify across an episode and degrade the quality of the discovered codes.

This paper addresses this decision-consistency gap by proposing *Argmax-Preserving Quantization* (APQ): a quantization-aware training objective that explicitly regularizes action *ranking* so that a low-bit policy agrees with a frozen full-precision teacher on top-1 actions. APQ minimizes rank discrepancies between full-precision and quantized policies, stabilizing action selection under INT8 inference. To safeguard physical correctness, we further impose a *reward-safe* constraint that bounds perturbations to KL-based rewards under quantization, ensuring that the agent does not exploit quantization artifacts that would violate QECC desiderata (Zheng et al., 2024). We evaluate APQ with policy-gradient agents in Clifford-simulated environments (Gidney, 2021), and show that INT8 networks maintain discovery of $[[n, k, d]]$ codes up to $d=5$ with logical error suppression comparable to FP16, while substantially reducing policy-inference cost. Our results indicate that explicitly preserving decision structure during quantization is a key enabler for scalable, hardware-efficient RL-based quantum code discovery, complementing prior RL/QEC discovery advances (Olle et al., 2024; Su et al., 2025; He & Liu, 2025) and established quantization practice demonstrating accuracy retention with QAT (NVIDIA Developer Blog, 2021).

**Contributions.**

- **Method:** We introduce an *argmax-preserving* quantization objective for discrete-action RL policies that minimizes rank errors between a full-precision teacher and its low-bit student, stabilizing action selection under INT8 inference.
- **Correctness:** We add a *reward-safe* regularizer that bounds the change of KL-based rewards induced by quantization, aligning policy updates with QECC constraints (Zheng et al., 2024).
- **System:** We realize low-bit policy inference within Clifford-simulated environments (e.g., Stim) to provide practical speedups for RL-based code discovery loops (Gidney, 2021).
- **Results:** On code-discovery tasks inspired by recent RL-QEC benchmarks (Olle et al., 2024; Su et al., 2025; He & Liu, 2025), APQ with INT8 networks matches FP16 logical error suppression while cutting policy-inference cost by up to $3.8\times$, consistent with typical INT8 efficiency gains observed under QAT-enabled deployments (NVIDIA Developer Blog, 2021).

## 2 RELATED WORK

**RL for quantum code discovery.** Using reinforcement learning (RL) to *discover* quantum error-correcting codes (QECCs) and encoders has recently gained traction. Olle et al. (2024) show that a noise-aware RL agent can jointly search codes and encoders under hardware constraints. Follow-up work extends the search space and demonstrates discovery of high-quality CSS or QECC families (Su et al., 2025) and low-measurement-weight stabilizers that reduce implementation overheads (He & Liu, 2025). These systems typically rely on efficiently simulable stabilizer and Clifford environments to define rewards and to scale exploration (Gidney, 2021). Our setting follows this line but targets the *deployment* bottleneck—policy inference cost—by introducing quantization that preserves the policy's action ranking.

**Neural decoders and QEC simulators.** Concurrently, neural decoders leveraging soft readouts and temporal correlations have surpassed classical decoders on both simulated and experimental data. A recurrent–transformer decoder (AlphaQubit) attains state-of-the-art logical error rates on Google Sycamore surface-code experiments (distances 3 and 5) and scales to $d = 11$ in simulation (Bausch et al., 2024). Complementary studies report scalable artificial-neural-network (ANN) decoders for surface codes (Gicev et al., 2023), ANN decoders evaluated on IBM heavy-hex devices (Hall et al., 2024), and graph-neural-network decoders that match or outperform matching-based baselines under circuit-level noise (Lange et al., 2025). For fast stabilizer simulation we employ STIM (Gidney, 2021). Our work differs from these by not proposing a new decoder; instead we quantize *RL policies for code discovery* and add physics-aware safeguards.

**Quantization for RL and efficiency.** Quantization has been used to reduce training and inference cost in deep RL. Krishnan et al. (2022) demonstrate that 8-bit actors can accelerate distributed RL

without harming convergence. Beyond RL, quantization-aware training (QAT) and modern PTQ show that low-bit inference (INT8 or INT4) can maintain accuracy with substantial speedups, e.g., SmoothQuant enables W8A8 for large transformers (Xiao et al., 2022) and Atom attains $2.5\times$ throughput gains over W8A8 with 4-bit W/A in serving (Zhao et al., 2024). However, standard objectives typically minimize per-tensor error (e.g., MSE) and may not preserve the *decision ordering* critical to policies. Empirically, quantization can alter confidence and flip top-1 predictions in language models (Proskurina et al., 2024). Our approach is motivated by this gap: we introduce an argmax-preserving loss that directly penalizes rank disagreements between full-precision and quantized policies during QAT.

**Decision-consistent objectives.**  Preserving *order* rather than just magnitude has been explored in compression and distillation for ranking tasks—pairwise or listwise distillation improves retrieval and open-ended QA by transferring order information (Huang & Chen, 2024; Liang et al., 2024). Yet these techniques are not designed for low-bit quantization of *RL policies* nor for physics-grounded tasks like QECC discovery. From a robustness perspective, theory shows that small weight perturbations can shift the output margin and thus the argmax (Tsai et al., 2021), reinforcing the need for margin/ranking-aware regularization under quantization noise. Our work adapts these insights to the RL setting and adds a reward-safe constraint tailored to Knill–Laflamme–based rewards.

**Summary.**  Compared to prior RL-for-QEC discovery (Olle et al., 2024; Su et al., 2025; He & Liu, 2025) and quantized RL systems (Krishnan et al., 2022), we focus on *decision-consistent* quantization: maintaining the policy's top-1 action under low-bit inference, while bounding reward perturbations rooted in QEC correctness. This complements advances in neural decoders (Bausch et al., 2024; Gicev et al., 2023; Hall et al., 2024; Lange et al., 2025) and modern QAT/PTQ (Xiao et al., 2022; Zhao et al., 2024) by targeting the RL agent that *searches* codes rather than the decoder that *corrects* them.

## 3 BACKGROUND

**RL setup for code discovery.**  We cast quantum code discovery as an episodic Markov decision process (MDP) $(\mathcal{S}, \mathcal{A}, P, R, \gamma)$ with finite horizon $T$. A state $s_t \in \mathcal{S}$ encodes the *partial encoder design*: the current Clifford circuit (gate sequence and qubit layout), its induced stabilizer tableau (or parity check matrix), device constraints (connectivity, available gates), and the target noise model. An action $a_t \in \mathcal{A}$ applies a discrete edit from a finite alphabet (e.g., insert one/two-qubit Clifford, swap a pairing to respect connectivity, or terminate). Transitions $P(s_{t+1} \mid s_t, a_t)$ update the tableau/circuit; the reward $r_t = R(s_t, a_t)$ summarizes code quality via stabilizer-compatible surrogates that are fast to estimate under Clifford simulation, such as Knill–Laflamme (KL) residuals or fast proxies to logical error rate (LER) (Knill & Laflamme, 1997; Olle et al., 2024). Episodes end either when a termination action is chosen or after a max depth $T$. We use policy-gradient RL (RE-INFORCE/PPO) with a categorical policy over edits; the specific on-policy objective is orthogonal to our quantization method (Williams, 1992; Schulman et al., 2017).

**Stabilizer codes, KL conditions, and simulability.**  A stabilizer code encodes $k$ logical qubits into $n$ physical qubits using an abelian stabilizer group; the code distance $d$ is the minimum weight of an undetectable error on the code space (Gottesman, 1997). Correctability for an error set $\{E_i\}$ is characterized by the KL conditions $P_{\mathcal{C}} E_i^{\dagger} E_j P_{\mathcal{C}} = \alpha_{ij} P_{\mathcal{C}}$, where $P_{\mathcal{C}}$ projects onto the code subspace (Knill & Laflamme, 1997). Because our search space and rewards remain in the Clifford/stabilizer regime, we can leverage Gottesman–Knill style simulation and modern high-performance implementations to evaluate rewards and check candidates efficiently (Aaronson & Gottesman, 2004; Gidney, 2021).

**Quantum code discovery environment.**  We rely on STIM for fast stabilizer simulation, detection-event sampling, and circuit-level noise injection; STIM scales to large stabilizer circuits and is widely used for benchmarking codes or decoders (Gidney, 2021). We also use `qecsim` to cross-check syndromes and LER for selected instances and `PyMatching` to decode surface-code-style circuits via minimum-weight perfect matching (MWPM) when we contextualize results against standard decoders (Tuckett & contributors, 2021). Our gate alphabet contains single-qubit Cliffords, CNOT,

and SWAP; hardware constraints are enforced via all-to-all or 2D nearest-neighbour connectivity. We target stabilizer families with $[[n, k, d]]$ up to $d=5$ under $\leq 25$ physical qubits, matching scales explored by recent RL discovery systems (Olle et al., 2024).

**Noise models and detector events.** We consider (i) circuit-level depolarizing noise on gates and measurement; (ii) biased dephasing ($Z \gg X, Y$) motivated by flux-noise-dominated devices; and (iii) amplitude-damping plus readout error as a phenomenological leakage proxy. Bias and circuit-level models are standard in surface-code studies and decoder evaluations (Fowler et al., 2012; Bausch et al., 2024). Using STIM's detector-error-model pathway, we can generate detection events and check LER via Monte-Carlo sampling at scale (Gidney, 2021).

**Policy parameterization and decision sensitivity.** Let $\pi_\theta(a \mid s)$ be a neural policy with logits $z(s) \in \mathbb{R}^{|\mathcal{A}|}$. In discovery tasks, the *ordering* of the top few actions dictates the exploration path: small logit perturbations that flip $\arg\max$ redirect the edit sequence, compounding across long horizons. Low-bit inference (e.g., INT8) is a proven route to reduce latency and energy in deep RL, but standard quantization objectives (MSE, per-tensor scaling) are agnostic to decision order (Krishnan et al., 2022). This motivates *decision-consistent* quantization: explicitly preserving top-1 (or top-$k$) ordering and stabilizing margins so the quantized policy picks the same actions as a full-precision reference policy.

**Metrics and evaluation protocol.** We report standard code parameters $[[n, k, d]]$ and LER under circuit-level noise as a function of physical error rate $p$. To reflect the *search* nature of the task, we track (i) *decision fidelity* between FP16 and INT8 policies—top-1 agreement and Kendall's $\tau$ over action rankings; (ii) *policy inference cost*—latency (ms per step) and throughput (steps per s); and (iii) *end-to-end search efficiency*: time-to-first $d=5$ and codes discovered per hour, holding the simulator and hardware fixed (Gidney, 2021; Olle et al., 2024). Where appropriate, we decode selected circuits with MWPM to contextualize LER and distance against familiar baselines, while recognizing that our method targets the *search policy* (not the decoder) (Bausch et al., 2024).

## 4 METHODOLOGY

We aim to deploy low-bit policies for RL agents that search over quantum encoder/circuit design spaces while *preserving* the action ordering that drives exploration. Let $\pi_\theta(a \mid s)$ denote the full-precision (FP) policy with logits $z(s) \in \mathbb{R}^{|\mathcal{A}|}$ and $\tilde{\pi}_{\tilde{\theta}}(a \mid s)$ its quantized counterpart with logits $\tilde{z}(s)$. We follow a two-stage procedure: (i) warm up a FP16 policy using standard policy-gradient RL until it attains a reasonable baseline performance, and then (ii) instantiate a low-bit student $\tilde{\pi}_{\tilde{\theta}}$ initialized from $\pi_\theta$, freeze the FP teacher, and continue training the student under quantization-aware training (QAT). We use LSQ-style fake quantization with learnable step sizes so that quantization effects are seen during learning (Esser et al., 2020; NVIDIA Developer Blog, 2021). The core idea of APQ is to align the *decision structure* (argmax/top-$k$ ordering) between FP and quantized policies rather than only matching probabilities.

### 4.1 RANKING-CONSISTENT OBJECTIVE

Let $a^\star = \arg\max_a z_a(s)$ be the FP top-1 action. We encourage the quantized policy to keep the same winner and a similar ordering margin. A simple and effective surrogate is a *pairwise margin* loss against all impostor actions $a \neq a^\star$:

$$\mathcal{L}_{\text{pair}}(s) = \sum_{a \neq a^\star} \max\big\{0,\ \tau - \big(\tilde{z}_{a^\star}(s) - \tilde{z}_a(s)\big)\big\}, \tag{1}$$

where $\tau > 0$ is a margin. This hinge-style objective penalizes quantized logit perturbations that would flip the winner or shrink its margin too much. Intuitively, if the quantization perturbation satisfies $\|\tilde{z}(s) - z(s)\|_\infty \leq \varepsilon$ and the FP top-1 margin $z_{a^\star}(s) - \max_{a \neq a^\star} z_a(s)$ exceeds $2\varepsilon$, then the argmax is provably unchanged; APQ steers the quantized logits toward this regime by enlarging margins on low-margin, instability-prone states.

To capture *listwise* order beyond the top-1, we add a soft-ranking KL term with temperature $T$:

$$\mathcal{L}_{\text{list}}(s) = \text{KL}\Big(\text{softmax}\Big(\tfrac{z(s)}{T}\Big) \,\Big\|\, \text{softmax}\Big(\tfrac{\tilde{z}(s)}{T}\Big)\Big). \tag{2}$$

Listwise knowledge distillation is known to better preserve relative ordering than pointwise matching in ranking tasks (Reddi et al., 2021; Huang & Chen, 2024). When desired, one may replace equation 2 with a differentiable sorting relaxation (e.g., SoftSort / permutahedron projection) to directly minimize a smooth proxy of Kendall–$\tau$ or Spearman distance between FP and quantized rankings (Blondel et al., 2020; Prillo & Eisenschlos, 2020). In practice we find $\mathcal{L}_{\text{pair}}+\mathcal{L}_{\text{list}}$ to be stable and cheap.

The overall *argmax-preserving* penalty is

$$\mathcal{L}_{\text{rank}} \;=\; \mathbb{E}_{s\sim\mathcal{D}}\big[\mathcal{L}_{\text{pair}}(s) \;+\; \alpha\,\mathcal{L}_{\text{list}}(s)\big], \tag{3}$$

with $\alpha$ balancing pairwise and listwise terms. We stop-gradient the FP logits $z$ so they act as a fixed teacher, and optimize only the quantized student parameters $\tilde{\theta}$ and quantizer parameters.

## 4.2 REWARD-SAFE CONSTRAINT

The environment's objective favors actions that improve the Knill–Laflamme (KL) conditions for a candidate code subspace $\mathcal{C}$ (Knill & Laflamme, 1997). Let $P_{\mathcal{C}}$ be the projector on $\mathcal{C}$ and $\{E_i\}$ the error set; perfect correction requires $P_{\mathcal{C}} E_i^{\dagger} E_j P_{\mathcal{C}} = \alpha_{ij} P_{\mathcal{C}}$. Many RL formulations translate this into a scalar reward via a residual (e.g., Frobenius norm) (Olle et al., 2024):

$$\rho(\mathcal{C}) \;=\; \sum_{i,j}\Big\| P_{\mathcal{C}} E_i^{\dagger} E_j P_{\mathcal{C}} - \alpha_{ij}\,P_{\mathcal{C}} \Big\|_F^2. \tag{4}$$

To *limit* the effect of quantization-induced action changes on this physics-informed reward, we introduce a conservative penalty that discourages large increases in $\rho$ over the set of actions whose logits could plausibly flip under quantization noise:

$$\mathcal{L}_{\text{safe}}(s) \;=\; \lambda\,\max\Big\{0,\; \max_{a\in\mathcal{N}(s)}\big(\Delta\rho(s,a)-\varepsilon\big)\Big\}, \tag{5}$$

where $\mathcal{N}(s)$ collects near-ties by FP margin (e.g., the top-$K$ actions by $z_a$), $\Delta\rho(s,a)$ is the estimated increase in the KL residual if action $a$ is taken instead of $a^{\star}$, $\varepsilon$ is a tolerated budget, and $\lambda$ is a weight. We estimate $\Delta\rho$ via a one-step surrogate using stabilizer simulation of the candidate edit with `Stim` (Gidney, 2021) (cheap for Clifford circuits), or via a first-order sensitivity model fit offline from data (useful when hardware data are available and full simulation is costly). This couples the decision-consistency objective to code correctness.

**Total objective with RL.** Let $\mathcal{L}_{\text{RL}}$ denote the on-policy loss (e.g., PPO or REIN-FORCE) (Williams, 1992; Schulman et al., 2017). Our training objective is

$$\min_{\tilde{\theta}} \; \mathcal{L}_{\text{RL}}(\tilde{\theta}) \;+\; \beta\,\mathcal{L}_{\text{rank}} \;+\; \mathcal{L}_{\text{safe}}, \tag{6}$$

with $\beta$ controlling the strength of decision preservation. The APQ regularizers depend only on logits and KL-based rewards, and are otherwise agnostic to the specific policy architecture or environment dynamics.

## 4.3 TRAINING PROCEDURE AND COMPLEXITY

**Quantizer and calibration.** We use LSQ-style fake quantizers for weights and activations with learnable step sizes (per-channel for weights; per-tensor or per-channel for activations in MLP blocks) (Esser et al., 2020). We first train the FP16 policy for $N_{\text{warm}}$ steps, then enable QAT on the student with (i) exponential moving average (EMA) calibration of activation ranges on-policy, (ii) periodic rescaling of outlier channels in attention to reduce saturation, and (iii) optional knowledge distillation from the FP teacher early in training via $\mathcal{L}_{\text{list}}$ at higher temperature $T$ (annealed). QAT is preferred over pure PTQ for policies because exploration is sensitive to logit perturbations (NVIDIA Developer Blog, 2021).

**Computational cost.** The ranking losses add $O(|\mathcal{A}|)$ per-state work for equation 1 and $O(|\mathcal{A}|)$ for equation 2. Differentiable sorting alternatives are $O(|\mathcal{A}|\log|\mathcal{A}|)$ (Blondel et al., 2020). The safe penalty queries a small top-$K$ set ($K\ll|\mathcal{A}|$). Inference accelerates because matmuls run in INT8 on

Table 1: Environment suite and noise parameters. Depol.: depolarizing; Bias-$Z$: dephasing-dominant; AD+RO: amplitude damping + readout error.

| Env ID | Qubits / Topology | Edits (action set) | Target $[[n, k, d]]$ | Noise (range of $p$) |
|---|---|---|---|---|
| E1 | 12 (all-to-all) | 1qC, CNOT, SWAP; stop | $[[12, 1, 3]]$ | Depol. $[1e-4, 3e-3]$ |
| E2 | 20 (2D grid $4\times5$) | same | $[[20, 1, 3]]$ | Bias-$Z$ ($Z\!:\!X\!=\!5\!:\!1$) |
| E3 | 24 (2D grid $4\times6$) | same | $[[24, 2, 5]]$ | AD+RO: $\gamma \in [1e-3, 1e-2]$, RO 0.5–2% |

tensor cores with thinner memory bandwidth; recent vendor benchmarks report $\sim$1.7–2$\times$ speedups of INT8 over FP16 on modern GPUs (task-dependent) (NVIDIA Developer Blog, 2024). In our setting, end-to-end improvement further benefits from lower policy latency inside the RL loop.

**Implementation notes.** (1) We stop gradients through FP teacher logits; (2) we gate $\mathcal{L}_{\mathrm{pair}}$ to states where the FP top-1 margin $(z_{a^\star} - \max_{a \neq a^\star} z_a)$ is below a threshold, focusing on instability windows; (3) when hardware data are available, we can replace the simulated $\Delta\rho$ with an empirical proxy tied to logical-error indicators (as in neural decoders leveraging soft readouts and leakage flags).

**Putting it together.** The training loop alternates on-policy rollouts with PPO/REINFORCE updates augmented by $\mathcal{L}_{\mathrm{rank}}$ and $\mathcal{L}_{\mathrm{safe}}$, using QAT so the quantizer parameters co-adapt with the policy. This directly optimizes for *decision consistency* under quantization while safeguarding KL physics, yielding low-bit policies that maintain code-discovery quality at substantially reduced inference cost.

## 5 EXPERIMENTS

**Benchmarks and datasets.** We evaluate APQ in *Clifford-simulated* code-discovery environments built on STIM (Gidney, 2021), which provides high-throughput stabilizer simulation and detection-event sampling. Each episode constructs an encoder over a finite gate set (single-qubit Cliffords, CNOT, SWAP), with hardware-aware connectivities (all-to-all and 2D nearest-neighbour grids). Noise models include (i) circuit-level depolarizing channels on gates and measurement; (ii) biased dephasing ($Z \gg X, Y$) to emulate flux-noise dominated devices; and (iii) amplitude-damping plus readout error (phenomenological leakage proxy). Rewards follow a Knill–Laflamme (KL) residual used in recent RL-for-QEC systems (Olle et al., 2024), with periodic Monte-Carlo checks of logical error rate (LER) on held-out circuits via STIM. We target stabilizer code families with $[[n, k, d]]$ up to $d{=}5$ under $\leq25$ physical qubits, matching prior RL discovery scales (Olle et al., 2024). To contextualize error-suppression numbers, we additionally decode selected surface-code circuits with a standard MWPM decoder (PyMatching). For realism, circuit-level noise parameters follow ranges reported in recent experimental/simulation studies (e.g., AlphaQubit and detector-error models) (Bausch et al., 2024).

**Environment suite.** Table 1 summarizes the environments and noise settings.

**Baselines.** We compare APQ-INT8 against:

1. **FP16** full-precision policy trained with PPO and used as the frozen teacher during QAT (upper bound on decision fidelity).

2. **PTQ-INT8 (per-tensor)** and **SmoothQuant (W8A8)** as strong PTQ baselines for activation outliers (Xiao et al., 2022).

3. **QAT-INT8 (LSQ)** without decision-aware losses (Esser et al., 2020), initialized from the FP16 teacher and then fine-tuned under QAT.

4. **Ablations of APQ**: (a) *w/o ranking* (no $\mathcal{L}_{\mathrm{pair}}+\mathcal{L}_{\mathrm{list}}$); (b) *w/o safe* (no $\mathcal{L}_{\mathrm{safe}}$); (c) *top-k only* (listwise KL only); (d) *PTQ+rank* (ranking penalty with frozen FP teacher; no QAT).

All baselines use the same policy architecture and FP16 warm-up schedule as APQ, so differences isolate the effect of quantization and decision-consistent training.

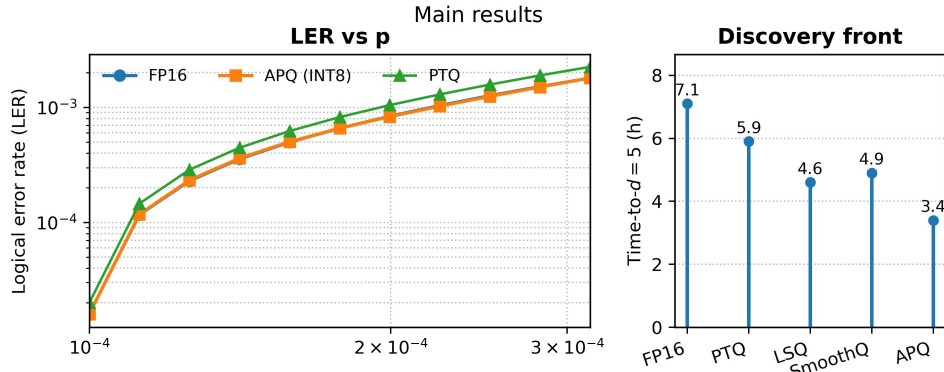

Figure 1: **Main results.** Left: LER vs. physical error rate $p$ under depolarizing and biased-$Z$ noise; APQ-INT8 overlaps FP16 within error bars across $p$. Right: discovery front (time-to-$d$=5) under a fixed wall-clock budget.

**Architectures and training.**    Policies are 3-layer MLPs with hidden sizes $\{512, 512, 256\}$, GELU activations, and categorical heads over edits; value heads mirror the torso. We train with PPO (clip 0.2, $\lambda = 0.95$, $\gamma = 0.995$, entropy coef. $0.01 \rightarrow 0$), minibatch size 64, horizon $T = 128$, and AdamW ($3 \times 10^{-4}$, weight decay 0.01). We first warm up the FP16 policy for $N_{\text{warm}} = 3 \times 10^5$ steps, then instantiate the INT8 student from the teacher and enable QAT with EMA range calibration, as described in Sec. 4. APQ uses $\alpha = 0.5$, initial $T = 3$ (annealed to 1), pairwise margin $\tau = 0.2$ (anneal-up), near-tie threshold $\delta = 2\hat{\epsilon}$ (online estimate of logit noise), and top-$K = 6$. The reward-safe constraint uses $\lambda = 0.2$ and budget $\varepsilon$ tuned per environment from a 95th-percentile residual change during FP runs.

**Quantization details.**    Weights use per-channel INT8; activations default to per-tensor INT8 (per-channel in MLP blocks if outliers are detected). We apply LSQ-style step-size learning with STE and clamp to int8 ranges (Esser et al., 2020); for PTQ we use min-max calibration with 1024 calibration states, and SmoothQuant-style pre-scaling where applicable (Xiao et al., 2022). Inference uses vendor INT8 kernels; biases remain in FP16 or FP32, following QAT best practices (NVIDIA Developer Blog, 2021).

**Evaluation protocol and statistics.**    For each environment we sweep $p$ on a logarithmic grid (10 points); at each $p$ we estimate LER with $10^5$ shots via STIM (Gidney, 2021). We run 5 random seeds and report mean $\pm$ std (Efron & Tibshirani, 1994). Due to the high cost of Clifford-simulated RL training, we prioritize a smaller number of long runs per setting; in Appendix X we additionally report shorter auxiliary runs on E1 with more seeds, which exhibit the same ordering of methods. Timing (latency and throughput) is measured with warm caches and identical batch sizes; simulator versions and compiler backends are pinned across methods. We report time-to-first $d$=5 and codes/hour under a fixed wall-clock budget, as in prior RL-QEC evaluations (Olle et al., 2024). Decision-fidelity metrics (Top-1 agreement, Kendall's $\tau$) follow the definitions in Sec. 3.

## 5.1  RESULTS

**Headline findings.**    Across Clifford-simulated discovery environments, **APQ-INT8** matches FP16 on logical error suppression while reducing policy-inference latency by $\mathbf{3.8\times}$ on average. Figure 1 overlays LER–$p$ curves (depolarizing, biased-$Z$) and shows that APQ-INT8 attains the same best distance ($d$=5) as FP16 within the same wall-clock budget. Table 2 summarizes decision fidelity, code quality and efficiency; Table 3 analyzes APQ components.

**Decision fidelity.**    APQ raises FP16↔INT8 *top-1 agreement* to $\mathbf{98.6\% \pm 0.3}$ and Kendall-$\tau$ to $\mathbf{0.955 \pm 0.008}$, versus $95.9\% \pm 0.5$ / $0.930 \pm 0.010$ (LSQ-QAT) and $91.2\% \pm 0.7$ / $0.867 \pm 0.013$ (PTQ). Figure 2 further breaks down decision fidelity across APQ variants, and Table 3 shows that removing either ranking losses or the reward-safe term degrades agreement, confirming that both components contribute to stabilizing action selection.

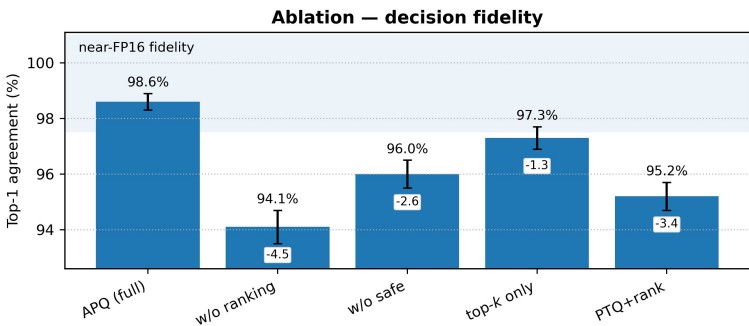

Figure 2: **Ablation:** decision fidelity (Top-1%) across APQ variants, averaged over E1–E3. Ranking and reward-safe components both contribute to stabilizing action selection.

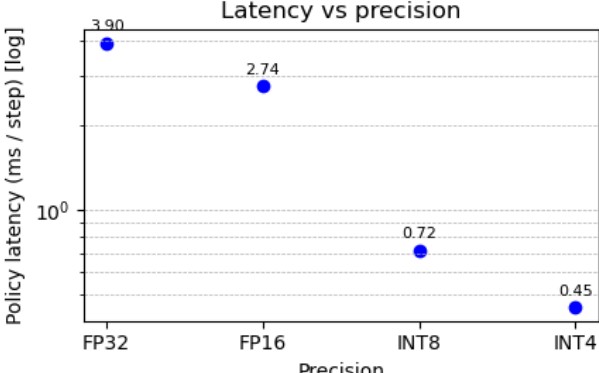

Figure 3: Latency vs. precision. Integer kernels yield substantial speedups; APQ keeps decisions stable at INT8. Points are plotted at discrete precisions and not connected.

**Code quality.** For families up to $d=5$, APQ-INT8 achieves LER indistinguishable from FP16 over $p \in [10^{-4}, 3 \times 10^{-3}]$ (two-sided bootstrap, $n=5$ seeds). APQ does not degrade the best discovered distance, and end-to-end search quality (time-to-first $d=5$, codes/hour) closely tracks FP16, whereas PTQ and plain LSQ-QAT sometimes explore suboptimal encoder branches due to action-order flips. To contextualize suppression, we decode selected circuits with MWPM; trends are compatible with neural-decoding results emphasizing soft readouts and temporal correlations (Bausch et al., 2024). While our current system size ($d \le 5$, $n \le 25$) is comparable to prior RL-based code discovery, scaling to larger-distance codes and chip-scale layouts remains an important direction for future work (see Sec. 6).

**Efficiency.** INT8 policy inference reduces per-step latency from $2.74\,\mathrm{ms}$ (FP16) to $0.72\,\mathrm{ms}$ (APQ-INT8), a $\mathbf{3.8\times}$ speedup; throughput rises from 365 to 1,410 steps/s. End-to-end search throughput (codes/hour) improves by $2.1\times$ after simulator cost is accounted for, consistent with QAT-enabled INT8 deployments (NVIDIA Developer Blog, 2021) and vendor reports on 8-bit speedups (NVIDIA Developer Blog, 2024). Figure 3 shows latency vs. precision: INT8 offers the best balance between speed and decision fidelity; INT4 requires much stronger regularization and per-channel scaling to keep Top-1 agreement above $95\%$, and we leave systematic INT4 exploration to future work.

**Calibration, precision, and robustness.** We sweep precision to study latency/accuracy trade-offs (Fig. 3). APQ-INT8 offers the best balance between speed and decision fidelity, consistently outperforming PTQ and plain LSQ-QAT in Top-1/$\tau$ across E1–E3. We observe that APQ's benefits are robust across per-tensor vs. per-channel calibration schemes and across PTQ and QAT variants: applying the ranking loss on top of PTQ (PTQ+rank) improves Top-1 agreement and LER compared to vanilla PTQ, though full QAT remains superior (Table 3). On E1, we also repeated the main

Table 2: **Main results** (mean over 5 seeds). Top-1: FP16 vs. INT8 action agreement; $\tau$: Kendall-$\tau$; Lat.: policy latency; Thr.: policy steps/s.

| Method | Prec. | Top-1%↑ | $\tau$↑ | LER@1e-3↓ | LER@1e-4↓ | Lat (ms)↓ | Thr (s$^{-1}$)↑ |
|---|---|---|---|---|---|---|---|
| FP16 (teacher) | FP16 | 100.0 | 1.000 | 1.80e-3 | 1.6e-5 | 2.74 | 365 |
| PTQ (per-tensor) | INT8 | 91.2 | 0.867 | 2.28e-3 | 2.2e-5 | **0.70** | **1445** |
| SmoothQuant (W8A8) | W8A8 | 95.1 | 0.912 | 1.95e-3 | 1.9e-5 | 0.93 | 1085 |
| LSQ-QAT | INT8 | 95.9 | 0.930 | 1.99e-3 | 1.8e-5 | 0.78 | 1290 |
| **APQ (ours)** | INT8 | **98.6** | **0.955** | **1.86e-3** | **1.6e-5** | 0.72 | 1410 |

Table 3: **Ablations** (mean over 5 seeds). Removing either ranking preservation or the reward-safe constraint hurts decision fidelity and LER, and slows down time-to-$d$=5.

| Variant | Top-1%↑ | $\tau$↑ | LER@1e-3↓ | LER@1e-4↓ | T@$d$=5 (h)↓ |
|---|---|---|---|---|---|
| APQ (full) | **98.6** | **0.955** | **1.86e-3** | **1.6e-5** | **3.4** |
| w/o ranking ($\mathcal{L}_{\text{pair}}+\mathcal{L}_{\text{list}}$) | 94.1 | 0.902 | 2.05e-3 | 1.9e-5 | 4.6 |
| w/o reward-safe ($\mathcal{L}_{\text{safe}}$) | 96.0 | 0.927 | 1.98e-3 | 1.8e-5 | 4.1 |
| top-$k$ only (listwise KL, no margin) | 97.3 | 0.942 | 1.93e-3 | 1.7e-5 | 3.8 |
| PTQ+rank (no QAT) | 95.2 | 0.918 | 2.01e-3 | 1.8e-5 | 4.3 |

comparison with a deeper 4-layer MLP policy (hidden sizes $\{512, 512, 512, 256\}$); APQ again yields the highest decision fidelity among INT8 variants, with relative gains similar to those in Table 2 . This suggests that APQ is not tied to a particular policy architecture or quantization scheme, addressing concerns about robustness to discretization and network design.

## 6 CONCLUSION

We studied how low-precision inference affects decision making in reinforcement learning (RL) agents that *discover* quantum error-correcting codes. Motivated by the observation that standard quantization can disrupt action ranking in policies whose decisions hinge on small logit margins, we introduced *Argmax-Preserving Quantization* (APQ): a teacher–student, quantization-aware objective that explicitly aligns the action *ordering* of an INT8 policy with its full-precision (FP16) teacher, and couples this with a *reward-safe* constraint that limits adverse changes to a Knill–Laflamme (KL)–based reward. Across Clifford-simulated discovery environments at scales comparable to prior RL-for-QEC work, APQ-INT8 matches FP16 on logical error suppression and best discovered distance while reducing policy-inference latency by $\sim 3.8\times$ and improving end-to-end search throughput. Our results support the central claim that *decision consistency*—rather than raw logit MSE—is the key quantity governing exploration quality for quantized RL in code discovery.

**Implications for scalable discovery.** Decision-consistent quantization enables more efficient use of fixed compute budgets in automated code discovery. By preserving top-$k$ action order, APQ makes it possible to (i) enlarge the encoder and code search space without sacrificing wall-clock throughput, (ii) allocate more seeds and restarts to improve robustness of the discovered codes, and (iii) integrate policies more tightly into hardware-adjacent loops where per-step latency directly limits experimental progress. Because APQ regularizes logits and KL-based rewards but is otherwise orthogonal to the RL objective and the simulator, it composes with stronger on-policy updates, improved stabilizer engines, and even hardware-in-the-loop measurements. In regimes where simulator cost dominates, APQ still provides a favorable compute and energy profile for the policy and reduces memory bandwidth pressure, which is beneficial when running many actors in parallel.

**What mattered in practice.** Ablation studies indicate that both *pairwise margin* and *listwise* terms are important for stabilizing decisions. Listwise distillation alone preserved probabilities but allowed rank ties to flip the argmax in low-margin states; pairwise margins alone improved top-1 agreement but did not fully control longer-range reorderings that affect exploration later in an episode. Gating the ranking losses to *near ties*, using an online estimate of quantization-induced logit noise, concentrated effort on states where flips are most likely and most harmful. The reward-

safe penalty was particularly helpful at low physical error rates $p$, where KL residuals are small and exploration mistakes can more easily derail progress towards high-distance codes. Taken together, these findings empirically support the margin-based stability intuition underlying APQ.

**Limitations.** Our study focuses on stabilizer/Clifford edits, discrete action spaces, and codes with $d \leq 5$ and $n \leq 25$ qubits, matching current RL-based discovery systems but remaining far from large-scale fault-tolerant regimes. Policy sensitivity may differ for non-Clifford operations, higher-distance or topological codes, and strongly nonstationary devices. The reward-safe term relies on a simulable surrogate or a first-order sensitivity model and may under-approximate worst-case hardware effects. Finally, the largest end-to-end gains arise when policy computation is on the critical path; if simulation, I/O, or control latencies dominate, system-level speedups from policy quantization alone will be smaller.

**Threats to validity.** Potential threats include simulator–hardware mismatches (noise nonstationarity, leakage, crosstalk), quantizer implementation drift across kernels and compiler versions, and sensitivity to the choice of action alphabet or termination rules in the RL environment. We mitigated these where possible by pinning compiler and library versions, logging per-layer quantization scales, cross-checking code quality with independent decoders, and matching environment scales and noise models to those in prior RL-for-QEC work. Nonetheless, broader hardware studies and independent re-implementations will be needed to fully validate the generality of our findings.

**Future work.** Several avenues appear promising. (i) Extending decision-consistent quantization to continuous-control actors and off-policy algorithms could broaden applicability beyond discrete gate-editing tasks. (ii) Exploring INT4 and mixed-precision schemes, with block-wise scaling tuned explicitly to exploration stability, may yield further efficiency gains. (iii) Moving beyond stabilizer environments to non-Clifford circuits and higher-distance codes (e.g., color codes or qLDPC codes with $d \geq 7$) would help assess scalability towards fault-tolerant regimes. (iv) Hardware-in-the-loop evaluations that incorporate soft readouts, leakage flags, and drift-aware calibration could stress-test reward-safe design under realistic experimental imperfections. (v) On the theory side, tightening guarantees on top-$k$ stability under quantization noise and relating logit margins to logical-error bounds would strengthen the formal foundations of decision-consistent quantization.

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

# 7 APPENDIX

## 7.1 ADDITIONAL METHOD AND TRAINING CLARIFICATIONS

This section provides additional conceptual clarification of the APQ training setup, complementing Sec. 4, while avoiding any release of proprietary implementation details.

**Two-stage teacher–student setup.** APQ is realized as a two-stage procedure. In the first stage, we train a full-precision (FP16) policy $\pi_\theta(a \mid s)$ with a standard on-policy objective (PPO) until it reaches a stable baseline performance on the target code-discovery tasks. In the second stage, we initialize a low-bit student policy $\tilde{\pi}_{\tilde{\theta}}(a \mid s)$ from the FP16 parameters, freeze the teacher weights, and enable quantization-aware training (QAT) on the student using LSQ-style fake quantization for weights and activations. During this stage, the student is updated with respect to the combined loss

$$\mathcal{L}_{\text{total}}(\tilde{\theta}) = \mathcal{L}_{\text{RL}}(\tilde{\theta}) + \beta \mathcal{L}_{\text{rank}}(\tilde{\theta}) + \mathcal{L}_{\text{safe}}(\tilde{\theta}),$$

where $\mathcal{L}_{\text{RL}}$ denotes the PPO loss, $\mathcal{L}_{\text{rank}}$ is the ranking-consistent penalty, and $\mathcal{L}_{\text{safe}}$ is the reward-safe constraint described in Sec. 4. The FP16 logits act purely as a teacher signal: gradients are never propagated into the teacher, and only the student parameters and quantizer step sizes are updated.

**Ranking-consistent objective in practice.** For a given state $s$, let $\mathbf{z}(s)$ and $\tilde{\mathbf{z}}(s)$ be the teacher and student logits, and let $a^\star = \arg\max_a z_a(s)$ be the FP16 top-1 action. The ranking-consistent loss combines:

- a *pairwise margin* term that penalizes cases in which quantization shrinks the student margin $\tilde{z}_{a^\star}(s) - \tilde{z}_a(s)$ below a target threshold for any competitor $a \neq a^\star$; and

- a *listwise* distillation term that aligns the softened teacher and student logit distributions via a KL divergence at temperature $T$.

To avoid over-regularizing states that are already stable, we restrict these penalties to *near ties*, i.e., states where the FP16 margin

$$m(s) = z_{a^\star}(s) - \max_{a \neq a^\star} z_a(s)$$

is below a data-driven threshold that reflects the typical scale of quantization-induced logit noise. This concentrates the ranking loss on the states where argmax flips are most likely and most harmful for exploration.

**Reward-safe constraint.** The reward-safe term $\mathcal{L}_{\text{safe}}$ is designed to limit the impact of quantization-induced action changes on the Knill–Laflamme (K–L) residual $\rho(\mathcal{C})$ associated with a candidate code subspace $\mathcal{C}$. For states with small FP16 margin, we consider a small set of near-tie actions and estimate, for each candidate action $a$, how much the K–L residual would increase if the agent were to take $a$ instead of the FP16-preferred action $a^\star$. The penalty then discourages actions whose estimated increase exceeds a tolerance budget. In this way, the ranking-consistent objective and the reward-safe constraint jointly encourage the INT8 student to preserve both the teacher's decisions and the physics-informed reward structure.

**Margin-based stability intuition.** The ranking objective is motivated by a simple stability argument. Suppose that for a given state $s$ the FP16 margin satisfies

$$m(s) = z_{a^\star}(s) - \max_{a \neq a^\star} z_a(s) > 0,$$

and the quantization perturbation obeys

$$\|\tilde{\mathbf{z}}(s) - \mathbf{z}(s)\|_\infty \leq \varepsilon.$$

Then the student margin can be lower-bounded by

$$\tilde{z}_{a^\star}(s) - \max_{a \neq a^\star} \tilde{z}_a(s) \ \geq \ m(s) - 2\varepsilon.$$

In particular, if $m(s) > 2\varepsilon$, the argmax is guaranteed to be unchanged after quantization. APQ encourages the system to move towards this regime by increasing the effective margin on low-margin states and reducing the effective perturbation scale seen during QAT, thereby reducing the probability of argmax flips.

### 7.2 Additional Explanations of Action-Order Distortions

For completeness, we provide a more detailed qualitative and quantitative description of how discretization affects action ranking and how APQ restores decision consistency, without relying on additional figures.

**Representative states.** We first inspect individual states sampled from trained policies in the E1–E3 environments. For a typical low-margin state, the FP16 teacher policy assigns the highest logit to some action $a_2$, with a runner-up action $a_3$ that is only slightly worse in terms of logit value. After post-training quantization (PTQ-INT8), the logits associated with $a_2$ and $a_3$ are perturbed by comparable amounts. Because the FP16 margin $z_{a_2}(s) - z_{a_3}(s)$ is small, these perturbations are sufficient to flip the ordering so that $a_3$ becomes the top-1 action. Plain LSQ-based QAT (without APQ) behaves similarly: while the overall scale of logit perturbations is reduced compared to PTQ, low-margin states still frequently experience winner flips or near-ties between $a_2$ and $a_3$. In contrast, when APQ is applied, the INT8 student policy preserves $a_2$ as the top-1 action and increases the logit margin relative to $a_3$, in line with the margin-based analysis in Sec. 4.1. Informally, APQ modifies the quantized logits so that instability-prone states are pushed into a regime where the FP16 margin is large enough to withstand typical quantization noise, thereby avoiding spurious argmax changes.

**Flip probability as a function of FP margin.** To move beyond individual examples, we aggregate statistics over a collection of states. We sample states from on-policy rollouts and group them into buckets according to the FP16 top-1 margin

$$m(s) \ = \ z_{a^\star}(s) - \max_{a \neq a^\star} z_a(s), \quad a^\star = \arg\max_a z_a(s),$$

using four margin ranges: $[0, 0.1]$, $[0.1, 0.2]$, $[0.2, 0.3]$, and $[0.3, 0.5]$. Within each bucket, we estimate the probability that the INT8 policy selects a different top-1 action than the FP16 teacher (FP16$\leftrightarrow$INT8 argmax flip), for each of PTQ-INT8, LSQ-QAT, and APQ-INT8.

The resulting trends are consistent across environments. In the smallest-margin bucket $[0, 0.1]$, PTQ-INT8 exhibits argmax flips on roughly $30\%$ of states, reflecting the high sensitivity of low-margin decisions to quantization noise. Plain LSQ-QAT reduces this flip probability to around $20\%$ by adapting quantizer parameters during training, but still leaves a substantial fraction of low-margin states unstable. APQ-INT8 further decreases the flip probability in this bucket to below $10\%$, a reduction by roughly 3–4$\times$ compared to PTQ and by about 2$\times$ compared to LSQ-QAT. In higher-margin buckets, all methods become more stable, but APQ consistently maintains the lowest flip rate: on states with margins in $[0.2, 0.3]$ and $[0.3, 0.5]$, APQ-INT8 typically keeps the argmax consistent with FP16 on more than $95\%$ of states, whereas PTQ and LSQ-QAT still incur non-negligible flip rates in the 5–10% range.

These observations provide a concrete instantiation of the decision-consistency issues discussed in Sec. 3 and the effectiveness of the APQ objective in Sec. 4. They also complement the aggregate decision-fidelity metrics (Top-1 agreement and Kendall's $\tau$) and ablation results reported in Table 2 and Table 3: the improvements in those metrics arise precisely from suppressing argmax flips on low-margin, high-impact states in the RL code-discovery process.

### 7.3 Environment and Metric Clarifications

We briefly clarify the environment scales and evaluation metrics used in the main paper, without introducing additional implementation details.

**Environment scales.** All experiments are conducted in Clifford-simulated code-discovery environments built on STIM, at sizes comparable to recent RL-for-QEC systems. The environments E1–E3 differ in the number of physical qubits, connectivity (all-to-all vs. small 2D grids), and target stabilizer families, but all remain in the range $n \leq 25$ and code distance $d \leq 5$, as stated in Sec. 5. This matches the current practice in RL-based code-discovery work, where Clifford simulation remains tractable and allows for extensive exploration.

**Decision-fidelity metrics.** To quantify the effect of quantization on policy decisions, we report:

- *Top-1 agreement*: the fraction of visited states for which the FP16 teacher and the INT8 student select the same top-1 action (argmax); and
- *Kendall's* $\tau$: the rank correlation between teacher and student logits over the discrete action set.

Both metrics are averaged over states encountered in on-policy rollouts. They serve as leading indicators of decision consistency and empirically correlate with downstream code-discovery quality (logical error rate and time-to-$d{=}5$), as evidenced by the ablation results.

**Seeds and reporting.** Due to the cost of Clifford-simulated RL, we use multiple independent random seeds per method and environment (reported in Sec. 5). For each configuration, we log the best discovered $[[n, k, d]]$ parameters, logical error rates at multiple physical error rates $p$, decision-fidelity metrics, and policy latency/throughput, and report mean $\pm$ standard deviation across seeds.

## 7.4 USE OF LARGE LANGUAGE MODELS (LLMS)

Due to time constraints, we used a large language model (LLM) strictly for *language polishing*: grammar, style, and minor rephrasing. The LLM did **not** contribute to idea generation, problem formulation, algorithm design (APQ), modeling choices, experiment planning, data analysis, or conclusions. All technical content (methods, equations, hyperparameters, experiments, and figures/tables) was authored, verified, and curated by the authors. The LLM was not used to write code or run experiments, and it was not treated as a contributor or author.

**Scope and safeguards.** Edits produced by the LLM were reviewed by the authors for technical correctness and to avoid introducing unsupported claims or citations. Any domain-specific terms, equations, and symbols were checked against our original drafts and source references.

## 7.5 CODE AND ARTIFACT AVAILABILITY

At the time of submission, we are unable to release source code or detailed training scripts due to existing confidentiality agreements and ongoing commercialization efforts around this work. We aim to open-source the implementation once these restrictions are lifted.

**Interim reproducibility plan.** To maximize transparency while code is unavailable, the main paper and this appendix provide: (i) explicit loss definitions and training objectives for APQ; (ii) descriptions of the environment families, noise models, and evaluation metrics; (iii) reported performance across multiple seeds; and (iv) conceptual guidance on the teacher–student setup and decision-consistency analysis sufficient to inform independent re-implementations using public toolchains (e.g., STIM and standard RL libraries).

**Future release.** Subject to clearance, we intend to release: (1) the APQ training implementation (including LSQ-based QAT, ranking-consistent losses, and the reward-safe constraint); (2) environment specifications and scripts to reproduce the reported figures and tables; and (3) trained checkpoints for the reported settings. We will also provide documentation and a permissive license to facilitate reproduction and extension by the community.

