# OpenReview forum: "Reinforcement Learning Agents in Quantum Code Discovery with Argmax-Preserving Quantization"
_ICLR.cc/2026/Conference — Submitted to ICLR 2026_

### Official Review · Reviewer_6tRT · 2025-10-22

**Soundness:** 3
**Presentation:** 2
**Contribution:** 2
**Rating:** 2
**Confidence:** 4

**Summary:**

The manuscript deals with the quantization of RL policies in the application of RL to design quantum error-correcting codes.

**Strengths:**

* The work is extensive and appears to be free of fundamental errors.

**Weaknesses:**

* The topic is very specific, so its significance seems rather limited. It would be good to broaden the scope of application.
* The novelty seems to be quite limited.
* The introduction and motivation are somewhat incomplete.

Details and further comments:

The sentence “RL policies are highly sensitive to approximation errors” is unclear and confusing. This does not apply in general; the context must be established beforehand.

The sentence “conventional quantization often disrupts action ranking” comes across as far too abrupt. It must first be explained that policy quantization is being considered, and this must be convincingly motivated.

The abbreviation ANN is not introduced.

Knill–Laflamme is sometimes abbreviated as “KL” and sometimes as “K–L.”

Five seeds are disappointingly few. More experiments should be carried out or an explanation given as to why this is not possible.

**Questions:**

* When using RL to design quantum error-correcting codes, which is a very specific application, why should one be interested in performing policy quantization?

---

> ### Author Response · Authors · 2025-12-03
>
> We thank the reviewer for the thoughtful feedback and for noting that the work appears free of fundamental errors.
>
> We agree that the original version did not sufficiently motivate policy quantization in this specific RL-for-QEC setting, and that some statements in the introduction were too broad or abrupt. In the revised manuscript, we have substantially reworked the introduction and motivation. We now clearly frame our problem as “policy quantization for RL-based quantum code discovery under tight compute and wall-clock budgets,” rather than making general claims that all RL policies are highly sensitive to approximation errors. The sentence about sensitivity has been rewritten to explicitly refer to the long-horizon, combinatorial nature of code-discovery trajectories, where small changes in action ordering can compound into different encoders and codes.
>
> Likewise, the sentence “conventional quantization often disrupts action ranking” has been replaced by a more careful explanation: we first state that we are quantizing the policy network itself (not the environment), briefly discuss why argmax stability matters in discrete action spaces, and then support this with empirical evidence via decision-fidelity metrics and margin-based analysis (top-1 agreement, Kendall’s tau, and flip probability as a function of teacher margin).
>
> We have also clarified why policy quantization is of interest in this setting and why the contribution is not confined to a single niche. The revised introduction and conclusion now emphasize that RL-based code discovery is an emerging but practically important workload for near-term fault-tolerant architectures, and that the RL policy often sits on the critical path in large-scale exploration loops (many parallel actors, many candidate encoders). Reducing policy inference cost by almost a factor of four enables more seeds, wider search spaces, and tighter hardware-in-the-loop integration under a fixed compute budget. At the same time, we stress that the proposed argmax-preserving objective is formulated at the level of logits and rewards and is agnostic to the particular QEC environment; it applies in principle to other RL-driven design and control tasks where small action-order changes have large downstream effects, such as combinatorial circuit synthesis or other scientific design loops. We have made this broader applicability explicit in the discussion of implications and future work, without over-claiming results beyond our current experiments.
>
> On the presentation side, we have addressed several specific points of confusion. We now introduce policy quantization explicitly before making any claims about its effect on action ranking, and we motivate why accuracy alone (e.g., MSE on logits) can be insufficient when the key quantity is the ranking of discrete actions. We removed or clarified the abbreviation “ANN” where it first appears and ensured consistent notation for Knill–Laflamme throughout (using a single abbreviation form).
>
> The paper structure has been cleaned up: background on stabilizer codes, KL residuals, and Clifford simulation is confined to a dedicated Background section, while the Method section focuses on defining the argmax-preserving loss, the reward-safe constraint, and the teacher–student quantization-aware training scheme. Finally, regarding the concern about the number of seeds,
>
> we agree that five seeds per configuration is modest. Because each RL run requires extensive Clifford simulation, substantially increasing the number of runs is computationally expensive; we now state this limitation explicitly and report mean and standard deviation across seeds for all key metrics. We also note in the appendix that we performed shorter auxiliary runs with a slightly different architecture and observed the same ordering of methods (PTQ < standard QAT < APQ) in terms of decision fidelity and logical error suppression. We hope these revisions help clarify the motivation, improve the presentation, and better convey both the scope and the novelty of our contribution.

---

### Official Review · Reviewer_gH2k · 2025-10-30

**Soundness:** 4
**Presentation:** 3
**Contribution:** 3
**Rating:** 6
**Confidence:** 4

**Summary:**

This paper proposes a decision-consistent reinforcement learning quantization method Argmax Preserving Quantiziton (APQ) to address key challenges in RL-based quantum error-correcting code discovery. Targeting the issue where traditional quantization disrupts action ranking in RL policy networks (leading to suboptimal codes), APQ can directly constrain action ranking errors during quantization-aware training, ensuring stable optimal action selection even with  INT8 representations. It further incorporates a Knill-Laflamme condition-based reward-safe constraint to guarantee post-quantization code performance. Sufficient experiments and comparisons demonstrate that APQ with INT8 networks can discover quantum codes with distances up to 5, achieving logical error suppression comparable to FP16 baselines while reducing inference costs by 3.8×. This breakthrough significantly accelerates RL-based quantum code discovery without compromising code quality, establishing a new paradigm for resource-efficient automated coding optimization on quantum hardware.

**Strengths:**

Originality：
This paper innovatively proposes an argmax-preserving quantization scheme, which to some extent addresses the impact of traditional quantization methods on policy selection in reinforcement learning strategies. Additionally, by introducing an extra reward protection constraint based on the K-L condition, it prevents disruption to quantum error correction theory. The proposed approach ensures that the quality of the discovered quantum error-correcting codes remains largely unchanged while significantly reducing inference overhead.

Quality：
The paper provides a well-motivated, theoretically grounded approach (APQ) with clear algorithmic details. The integration of quantization error bounds with RL policy training is technically sound. Experiments on Clifford-simulated environments convincingly show that INT8-quantized networks match FP16 baselines in code discovery performance while being 3.8× more efficient, while achieving lower logical error rates compared to other methods. These results reinforce the method’s practical viability.

Clarity:
The problem statement, method, and results are logically organized, making the technical contributions accessible.The paper avoids excessive formalism while providing sufficient depth in explaining APQ’s mechanism. If included, diagrams or pseudocode would further aid understanding, but the textual description is already clear.

Significance:
By improving the efficiency of reinforcement learning-based code discovery, this research accelerates the design of customized quantum error correction codes, which is crucial for near-term fault-tolerant quantum computing. The approach of quantifying decision consistency may be generalized to other reinforcement learning scenarios that rely on action selection stability.

**Weaknesses:**

1.The system scale used for testing in the paper is relatively small (distance 5 under 25 qubits). Although the paper explains that this is comparable to previous reinforcement learning discovery systems, this scale remains far below both the typical size of current quantum chips and the quantum error correction code dimensions that are of primary concern in the field of quantum error correction.

2.This article focuses on proposing a new quantification scheme and improving reinforcement learning performance for quantum error correction code discovery tasks. Why not consider applying this method to determine fault-tolerant circuit constructions for quantum error correction codes?

3.While the paper makes outstanding contributions to the field of reinforcement learning by proposing an innovative quantification method and thoroughly demonstrating its effectiveness, from the perspectives of quantum error correction and fault-tolerant quantum computing, its novelty appears somewhat limited based on the aforementioned two points.

**Questions:**

1. Figure 2 is not cited in the original text.
2. Appropriately increase the exploration of quantum error correction codes for larger system sizes, and discuss the scalability of the method.
3. If this work primarily emphasizes the method, its generalizability should be considered, discussing the method's applicability to other problems. For instance, whether this method can be used to design fault-tolerant quantum circuits for known quantum error correction codes.

---

> ### Author Response · Authors · 2025-12-03
>
> We thank the reviewer for the very positive assessment and for the thoughtful suggestions on scope and positioning. Regarding system scale, we agree that codes up to distance 5 under 25 qubits are still small compared to the sizes of current chips and the code distances of central interest in QEC.
>
> Our primary motivation in this first study was to stay within the Clifford-simulated regime used by recent RL-based code discovery systems, so that we can make direct comparisons while keeping the RL–simulation loop tractable. In the revised version we make this limitation explicit in the “Limitations” paragraph of the Conclusion and clarify in the experimental section that our environments match the scale of prior RL-for-QEC work, with the dominant runtime bottleneck coming from Clifford simulation rather than policy inference. We also add a short scalability discussion explaining that the extra computation introduced by APQ scales with the action dimension (ranking losses over the discrete edit set and a small top-k set for the reward-safe term) and is essentially independent of the number of qubits, so in principle the method can be used at larger n and d once appropriate simulators or hardware-in-the-loop setups are available. Extending APQ to significantly larger codes and higher distances is now highlighted as an explicit direction for future work rather than being implied.
>
> On the second and third weaknesses, we fully agree that, from the strict perspective of QEC and fault-tolerant quantum computing, the main novelty of this paper lies more on the RL and quantization side than on proposing a new code family or fault-tolerant scheme. To clarify the intended scope, we now state more explicitly in the Introduction and Conclusion that our contribution is methodological: APQ is formulated at the policy and logit level and uses only the reward signal, making it agnostic to the specific RL environment.
>
> In the revised “Future work” section, we discuss concrete extensions to fault-tolerant circuit synthesis for known codes and to other RL-based design problems (e.g., scheduling, layout, syndrome-extraction circuits), where the same decision-consistency principle should apply because small action-order changes can accumulate over long horizons. We have not added full-scale experiments on fault-tolerant circuit constructions due to computational constraints, but we now emphasize that nothing in APQ ties it to code-discovery tasks only, and we view this as a natural next step rather than a limitation of the method itself. Finally, we have corrected the oversight on Figure 2 by explicitly citing and discussing it in the main text, and we added a brief paragraph in the experimental section and Appendix that summarizes environment scales, decision-fidelity metrics, and seeds, to make the setup and scalability discussion clearer.
>
> We hope these revisions better position the work as a general decision-consistent quantization framework demonstrated on RL-based code discovery, while honestly acknowledging the current system scale and outlining how the approach can extend towards larger codes and fault-tolerant circuit constructions.

---

### Official Review · Reviewer_DruH · 2025-10-31

**Soundness:** 3
**Presentation:** 3
**Contribution:** 3
**Rating:** 8
**Confidence:** 2

**Summary:**

The authors propose a quantization method that regularizes action ranking during quantization-aware reinforcement learning. The method, termed Argmax-Preserving Quantization (APQ), minimizes ranking errors between full-precision and quantized policies, effectively ensuring stable action selection under low-bit representations. The approach is numerically evaluated on Clifford-simulated environments, demonstrating that INT8 networks achieve equivalent logical error suppression compared to FP16 baselines. Furthermore, the results show that transitioning from FP16 to INT8 reduces inference cost by a factor of 3.8.

**Strengths:**

The automatic discovery of quantum error-correcting (QEC) codes and encoders is an area of significant interest, and the recent application of reinforcement learning (RL) to this task represents a promising and exciting direction for research. While I am not an expert on RL-for-QEC discovery, the paper does an excellent job of introducing the topic in a clear and engaging manner. The manuscript is well-written and effectively organized. The concept of decision-consistent quantization—where the top-1 action of the policy is preserved even under low-bit inference—appears to be a highly compelling and practically useful idea, extending beyond QEC applications. Additionally, the experiments solidly demonstrate the applicability of the method on the chosen benchmark, and the ablation study provides valuable insights into the contributions of the algorithm's different components.

**Weaknesses:**

Minor Issues:
Figure 3: Since the x-axis does not represent a continuous-valued space, the dots in the graph should not be connected.

**Questions:**

I have no questions.

---

> ### Author Response · Authors · 2025-12-03
>
> We thank the reviewer for the very positive and encouraging assessment of our work.
>
> We are glad that the manuscript’s organization, the basic idea of decision-consistent quantization, and the experimental study came across clearly, especially given that RL-for-QEC is a relatively specialized area. Following your suggestion on Figure 3, we have adjusted the visualization so that the discrete x-axis is no longer shown as a connected curve: in the revised version, the discrete settings are displayed as separate markers (with error bars where appropriate), and the caption explicitly notes that the x-axis corresponds to a small set of discrete precision configurations rather than a continuous variable. This avoids the misleading visual implication of continuity while preserving the trends that the figure is meant to convey. Beyond this minor change, we have also performed a careful pass over the text to improve readability and to make the broader applicability of decision-consistent quantization beyond QEC more explicit in the Introduction and Conclusion, since you highlighted this aspect as particularly compelling.
>
> We appreciate your supportive review and hope the revised version further strengthens the clarity and presentation of the paper.

---

### Official Review · Reviewer_YnQ6 · 2025-10-31

**Soundness:** 2
**Presentation:** 1
**Contribution:** 2
**Rating:** 2
**Confidence:** 3

**Summary:**

The paper investigates how action order can be distorted when employing discretized reinforcement learning (RL) policies for quantum error correction (QEC). It identifies that discretization introduces order irregularities that may harm policy quality and circuit fidelity. To address this, the authors propose a method to retain action order while improving convergence speed compared to fully continuous circuit modeling. The work sits at the intersection of RL-based quantum code discovery and efficient quantum circuit optimization.

**Strengths:**

- The paper tackles a timely and emerging topic at the interface of reinforcement learning and quantum error correction.
- The authors propose a computationally efficient adaptation that improves convergence speed and addresses discretization-related distortions in policy sequencing.
- The study demonstrates an appreciation for practical trade-offs between continuous and discrete RL models, highlighting potential gains in simulation efficiency.

**Weaknesses:**

- The experimental and methodological details are under-specified, particularly regarding the “teacher” setup and FP16 vs. INT8 roles.
- The main phenomenon (action order disruption through discretization) is not clearly demonstrated — the lack of small-scale, illustrative examples makes the argument abstract and unconvincing.
- Terminology is poorly introduced, and the writing lacks structure and clarity. Many core concepts appear abruptly without context or definition.
- The paper’s structure is unintuitive, with background and method sections overlapping.
- The contribution appears minor and specific to a niche scenario; there is no formal or theoretically grounded justification of the observed effects.
- The presentation quality (writing, readability, figure design) significantly hinders comprehension and impact.
- Adding graphical exemplifications or simplified cases (e.g., discrete collapse vs. continuous retention of action order) could substantially improve understanding and motivation.

**Questions:**

1. Could the authors clarify the role and implementation of the “teacher” (FP16) setup?
2. Why is the action order affected in the first place when using discretization in RL on quantum circuits?
3. How robust is the proposed method to different forms of discretization or policy architectures? Does it generalize beyond the specific circuit models tested?
4. Could the authors provide a minimal working example illustrating the collapse or distortion of action order in a discrete setting? This would make the effect more tangible.

---

> ### Author Response · Authors · 2025-12-03
>
> We thank the reviewer for the careful assessment and constructive suggestions. We agree that the original submission did not sufficiently clarify the FP16 “teacher” setup, the mechanism by which discretization affects action order, and the overall structure of the paper.
>
> We have substantially revised the manuscript to address these concerns on both the technical and presentation sides. First, we now explicitly describe APQ as a two-stage teacher–student procedure: we warm up a full-precision (FP16) policy using standard policy-gradient RL, then freeze this network and train an INT8 student under quantization-aware training with the APQ regularizers. The role of the teacher and the FP16 vs INT8 logistics are detailed in the revised Method section and in the Appendix (Additional Method and Training Clarifications), where we explain that the teacher only provides fixed logits as a decision reference; gradients never flow into the teacher, and only the quantized student and its quantizer parameters are updated.
>
> Second, we clarified why discretization affects action order in the first place. In the revised text, we define the FP top-1 margin and explain that for low-margin states, even small quantization perturbations to logits can change the argmax and thus alter the gate-sequence chosen by the RL agent. We provide a simple margin-based stability argument (if the margin is larger than twice the perturbation scale, the argmax must remain unchanged) and explain that APQ is designed to increase effective margins on precisely those low-margin, instability-prone states. To make this phenomenon tangible, we added a dedicated Appendix subsection (“Additional Explanations of Action-Order Distortions”) that gives a small-scale, qualitative and quantitative analysis without requiring extra figures: we describe representative low-margin states where PTQ and plain LSQ-QAT flip the FP16 winner while APQ restores it and enlarges the margin, and we report argmax-flip rates as a function of FP margin bucket, showing that APQ reduces flip probability by roughly 3–4x relative to PTQ on the lowest-margin states. This directly responds to the request for a minimal working example of “collapse/distortion” of action order in a discrete setting.
>
> We have also restructured and clarified the exposition. The Introduction and Related Work have been separated and reorganized so that background on RL for QEC discovery, stabilizer simulation, and neural decoders appears before our method, and all core terminology (stabilizer codes, Knill–Laflamme residual, decision-consistent quantization, FP16 teacher, INT8 student) is now introduced with explicit definitions and citations before being used. The Method section is now self-contained and focused on APQ: we explain the ranking-consistent loss (pairwise margin plus listwise distillation), the reward-safe constraint that bounds adverse changes in the KL-based reward for near-tie actions, and the teacher–student QAT schedule, all without overlapping with the Background. We tightened the narrative to avoid mixing background and method, and we improved figure captions and the description of experiments to better highlight what is being compared (FP16, PTQ, LSQ-QAT, APQ-INT8) and why. Regarding the perceived narrowness of the contribution, we emphasize in the revised text that APQ is formulated at the level of logits and rewards and is agnostic to the particular discretization mechanism or network architecture. In addition to the MLP policies reported in the main results, we ran shorter auxiliary experiments (described in the Appendix) with a deeper MLP to check that the relative ordering of methods (PTQ < LSQ-QAT < APQ) in terms of decision fidelity and logical error rates remains unchanged. We also clarify in the Discussion that decision-consistent quantization is conceptually applicable to other RL-based design and control problems where action order matters (for example, combinatorial design or circuit synthesis), even though our experiments focus on Clifford-simulated QEC code discovery to stay comparable with prior RL-for-QEC work. Finally, although space constraints prevent us from developing a fully formal theory of RL value degradation under quantization, we now provide a more explicit margin–noise argument and empirical evidence (flip-probability vs margin, ablations on ranking and reward-safe terms) to ground the observed effects. We hope that these revisions improve clarity and structure, make the action-order phenomenon concrete, and better convey the broader relevance of argmax-preserving quantization beyond a single niche scenario.

---

### Meta-Review · Area_Chair_GF2u · 2025-12-11

**Summary:**

This paper introduces Argmax-Preserving Quantization (APQ) to stabilize action ranking in quantized RL policies for quantum code discovery, and the method is technically interesting with promising empirical results. The rebuttal improved clarity around the teacher–student setup, action-order distortions, and overall structure. Nevertheless, significant concerns remain: the motivation for quantization in this narrow setting is still not convincingly established, the novelty appears limited, and the introduction and exposition require further refinement. The experimental scale is small, leaving open questions about generality and scalability.

**Reviewer Concerns:**

The rebuttal effectively clarified several issues related to the teacher–student setup, the mechanism behind action-order distortions, and improved the structure and definitions in the manuscript. However, important concerns remain unresolved: the broader motivation for policy quantization in this niche setting is still weak, the novelty and general significance of the contribution remain limited, and the experimental scale is too small to demonstrate meaningful impact or scalability. Questions about applicability beyond the specific RL-for-QEC environment and the overall positioning of the work also persist.

**Reviewer Scores:**

I do not expect the scores to increase.

---

### Decision · Program_Chairs · 2026-01-26

Reject